# *Usp22* Overexpression Leads to Aberrant Signal Transduction of Cancer-Related Pathways but Is Not Sufficient to Drive Tumor Formation in Mice

**DOI:** 10.3390/cancers13174276

**Published:** 2021-08-25

**Authors:** Xianghong Kuang, Michael J. McAndrew, Lisa Maria Mustachio, Ying-Jiun C. Chen, Boyko S. Atanassov, Kevin Lin, Yue Lu, Jianjun Shen, Andrew Salinger, Timothy Macatee, Sharon Y. R. Dent, Evangelia Koutelou

**Affiliations:** 1Department of Epigenetics and Molecular Carcinogenesis, University of Texas MD Anderson Cancer Center, Smithville, TX 78957, USA; xkuang@mdanderson.org (X.K.); mmcandrew@luminexcorp.com (M.J.M.); Lmustachio@mdanderson.org (L.M.M.); YChen382@mdanderson.org (Y.-J.C.C.); KeLin@mdanderson.org (K.L.); YLu4@mdanderson.org (Y.L.); jianshen@mdanderson.org (J.S.); apsalinger@mdanderson.org (A.S.); tlmacatee@mdanderson.org (T.M.); 2Center for Cancer Epigenetics, University of Texas MD Anderson Cancer Center, Houston, TX 77030, USA; 3Luminex Corporation, 12212 Technology Blvd. Suite 130, Austin, TX 78721, USA; 4Department of Pharmacology and Therapeutics, Roswell Park Comprehensive Cancer Center, Buffalo, NY 14263, USA; boyko.atanassov@roswellpark.org

**Keywords:** mammary gland epithelial cells, mouse, SAGA, *Usp22* overexpression, migration, PI3K-AKT, WNT-beta catenin, estrogen response early

## Abstract

**Simple Summary:**

Increased levels of the *Usp22* deubiquitinase have been observed in several types of human cancer, particularly highly aggressive, therapy-resistant tumors. However, the role of *Usp22* overexpression in cancer etiology is not known. To address whether *Usp22* overexpression is sufficient to induce tumors in vivo, we created a mouse that expresses high levels of *Usp22* in all tissues beginning in embryogenesis through to adulthood. Analyses of these mice clearly show that while *Usp22* overexpression alters several tumor-related signaling pathways and induces associated morphological changes such as hyperbranching of the mammary gland, *Usp22* overexpression alone is not sufficient to induce tumorigenesis. These findings indicate that *Usp22* likely enhances tumor formation and progression through cooperation with oncoproteins that exacerbate abnormal signal transduction.

**Abstract:**

*Usp22* overexpression is observed in several human cancers and is correlated with poor patient outcomes. The molecular basis underlying this correlation is not clear. *Usp22* is the catalytic subunit of the deubiquitylation module in the SAGA histone-modifying complex, which regulates gene transcription. Our previous work demonstrated that the loss of *Usp22* in mice leads to decreased expression of several components of receptor tyrosine kinase and TGFβ signaling pathways. To determine whether these pathways are upregulated when *Usp22* is overexpressed, we created a mouse model that expresses high levels of *Usp22* in all tissues. Phenotypic characterization of these mice revealed over-branching of the mammary glands in females. Transcriptomic analyses indicate the upregulation of key pathways involved in mammary gland branching in mammary epithelial cells derived from the *Usp22*-overexpressing mice, including estrogen receptor, ERK/MAPK, and TGFβ signaling. However, *Usp22* overexpression did not lead to increased tumorigenesis in any tissue. Our findings indicate that elevated levels of *Usp22* are not sufficient to induce tumors, but it may enhance signaling abnormalities associated with oncogenesis.

## 1. Introduction

Cancer is the result of somatic genetic and epigenetic lesions that lead to deregulated gene expression and subsequent abnormal cellular function. Aberrant gene expression and altered epigenetic profiles greatly contribute to the gradual transformation of normal cells into tumor cells through a multi-stage process [1]. Given the biochemical functions of *Usp22* as part of the SAGA complex in transcriptional activation and post-translational modification of both histone and non-histone targets, *Usp22* provides an intriguing target for the development of new therapeutic interventions. Overexpression of *Usp22* was initially correlated with increased tumor growth, metastatic risk, and death from cancer in transcriptomic studies investigating the role of factors that regulate stemness in prostate cancer [2,3]. Subsequent studies have shown that *Usp22* is overexpressed in many cancer types, including breast cancer, lung cancer, and pancreatic cancer [4]. However, how *Usp22* contributes to oncogenesis is not known.

Knock-down and overexpression studies in cancer cell lines suggest that *Usp22* may promote cancer through the positive regulation of well-known oncogenes including BMI1 [5], c-MYC [6], SIRT1 [7], cyclin B1 [8], and KDM1A [9], mainly through the regulation of cell cycle progression. *Usp22* has also been linked to the negative regulation of tumor suppressors such as p53 [7,10,11], again indicating connections to the control of proliferation regulation. Consistent with these findings, changes in *Usp22* levels impact the expression of genes regulated by the androgen receptor (AR) together with MYC in xenograft models for prostate cancer through the regulation of AR protein levels, ultimately affecting the response to therapy [12]. In a subsequent study, conditional overexpression of *Usp22* in mouse prostate lobes increased the proliferation of prostate epithelial cells, and transcriptomic analyses again identified changes in cell cycle genes upon altered expression of *Usp22*, as well as changes in genes involved in DNA repair. However, no hyperplasias or neoplastic formations were observed [13]. Interestingly, the nucleotide excision repair protein XPC was identified as a direct substrate of *Usp22*, and loss of *Usp22* sensitized cells to genotoxic insults [13]. These findings indicate that *Usp22* is “pro-oncogenic” and that it can impact the response to chemotherapies.

In contrast to these pro-oncogenic functions of *Usp22*, a recent study indicated that *Usp22* shows a context-dependent tumor suppressor function in colorectal cancer [14]. Conditional deletion of *Usp22* in a mutant APC (APC^1638N^) mouse model resulted in induced aggressive tumor growth with increased proliferation and angiogenesis. Deletion of *Usp22* in human colorectal cancer cell lines increased tumorigenic phenotypes and increased mTOR activity [14].

The context-specific outcomes of altered *Usp22* expression in cancer cell lines and mouse models highlight the need to define *Usp22* functions in normal cells and processes. Deletion of *Usp22* in mice leads to embryonic lethality [7,15]. Systematic phenotypic and molecular characterization of the embryos revealed that the absence of *Usp22* resulted in a marked attenuation of multiple signaling cascades at midgestation, including those mediated by receptor tyrosine kinases (RTKs) and TGFβ receptor, leading to defective placental vasculature and embryo death by E14.5 [15]. A number of cellular processes were deregulated in the placenta of the *Usp22* null mice, including the propagation of endothelial cells, differentiation and specification of perivascular cells, and cell migration, likely due to aberrant intercellular signaling.

To complement these deletion studies and to determine whether *Usp22* overexpression impacts RTK, TGFβ receptor, and other pro-oncogenic signaling pathways, we generated an unbiased mouse model with global overexpression of a *Usp22* transgene inserted at the ROSA26 locus. Ubiquitous overexpression of *Usp22* resulted in aberrant ductal side-branching in the mammary epithelium of nulliparous female mice. Deregulation of signaling cascades that impact the migration of mammary epithelial cells, including the activation of integrin, ER receptor, ERK/MAPK, and TGFβ pathways, were observed in mammary epithelial cells derived from the *Usp22* overexpressing mice. However, no mammary epithelial hyperplasias or spontaneous tumors were observed in any tissue of the *Usp22* overexpressing mice. Taken together, our data further demonstrate direct links between *Usp22* overexpression and aberrant signaling in normal tissues, and they reveal a previously uncharacterized role for *Usp22* in mammary gland branching. Our work also clearly shows that *Usp22* overexpression alone is not sufficient to induce tumorigenesis.

## 2. Results

### 2.1. Generation of Overexpressing Usp22 Mice

We generated a conditional mouse model bearing a transgene for the overexpression of *Usp22* in order to determine the effects of upregulation of this deubiquitinase in vivo. To avoid the positional effects of random integrations in the mouse genome, *Usp22* integration was targeted to the ROSA26 locus. Mouse *Usp22* cDNA expression is driven by the strong CAGGS promoter, but a “STOP cassette” flanked by LoxP sites (LSL) prevents expression (Appendix A). Upon exposure to the Cre recombinase, the “STOP cassette” is removed and *Usp22* is expressed. The transgene construct also includes sequences encoding FLAG and MYC affinity tags in frame with *Usp22* and homology arms (3′ and 5′) for insertion into the ROSA26 locus (Figure 1A). We confirmed correct targeting into mouse embryonic stem (ES) cells by Southern blotting and DNA sequencing (Appendix A and data not shown). Chimeric mice were then generated by blastocyst injection of ES cells. The chimeric mice were backcrossed to wild-type mice to generate mice heterozygous for the *Usp22* insertion allele (*Usp22^LSL/+^).* For simplicity, we will refer to these mice as *LSL/+* (Figure 1A,B and Appendix A). Intercrosses of the heterozygotes gave rise to homozygous *LSL/LSL* mice.

The conditional allele allows controlled temporal and spatial induction of *Usp22* expression upon exposure to specific Cre recombinase transgenes that remove the STOP cassette. We decided first to use a *ZP3-Cre* transgene to induce ubiquitous overexpression (OE) of *Usp22* (Figure 1A) [16]. We confirmed overexpression in both heterozygous and homozygous OE mice (Figure 1B,C), which were born at expected Mendelian ratios (Table 1). Immunoprecipitation using an anti-FLAG antibody, followed by immunoblotting with an anti-MYC tag antibody confirmed the expression of the tagged *Usp22* protein in the liver and spleen of *OE/+* and *OE/OE* mice, and the lack of expression prior to excision of the STOP cassette in the *LSL/+* mice (Figure 1B). Expression of the tagged allele was also confirmed in the mammary gland, the cerebellum, kidney, liver, lung, and skin of the OE mice (Figure 1C,D and data not shown). *Usp22* mRNA levels, measured by qRT-PCR, indicated a 10–20 fold increase in expression in liver, spleen, and lung tissues dissected from homozygous (*OE/OE*) mice relative to their WT littermates (Figure 1C). To determine whether the tagged *Usp22* protein is assembled into the SAGA complex, we performed anti-FLAG immunoprecipitations followed by immunoblotting with ATXN7L3 and GCN5 antibodies. Both proteins were detected, indicating that FLAG-Usp22 associates with both the DUB and HAT modules of SAGA (Figure 1E). Interestingly, the levels of endogenous GCN5 and ATXN7L3 proteins in different tissues were not affected by the exogenous overexpression of *Usp22*, indicating that much of the overexpressed *Usp22* protein is likely not incorporated into the DUB module of the SAGA complex (Figure 1F). The tagged, overexpressed *Usp22* protein may compete with endogenous *Usp22* for incorporation into the SAGA complex and lead to a dominant negative phenotype if the affinity tags affect *Usp22* functions. The birth of OE/OE mice at expected Mendelian frequencies suggests that the overexpressed protein is functional (Table 1), but in order to further confirm the full functionality of the tagged *Usp22* protein, we crossed the *Usp22* OE mice with mice bearing a null allele of *Usp22*. Our previous studies and those of others [7,15] demonstrated that homozygous *Usp22* null mice die as embryos at midgestation. However, this lethality is completely rescued by one or two copies of the *OE* allele. Both OE/+; *Usp22*
^−/−^ and OE/OE; *Usp22*^−/−^ mice were born at expected Mendelian ratios and survived to adulthood (Table 2). Altogether, these data indicate that the overexpressed tagged *Usp22* protein is fully functional in vivo.

### 2.2. Mammary Gland Over-Branching in Usp22 Overexpressing Mice

We performed anatomical and histological analyses of tissues from adult *Usp22 OE/OE* mice to determine if *Usp22* overexpression resulted in specific phenotypic changes. One prominent phenotype we observed was hyperbranching of nulliparous female mammary glands (Figure 2A). To determine the time of onset of this phenotype, we performed whole-mount staining of mammary glands from WT and *Usp22 OE/OE* females at 3 weeks, 8 weeks, and 16 weeks of age. No measurable changes in branching were observed until 16 weeks, when side branching was significantly increased in the mammary glands of *Usp22*
*OE/OE* mice compared with WT littermates (Figure 2A,B). We did not observe overt increases in cell proliferation in the mammary gland as shown by immunohistochemistry performed against phospho-H3 (Figure 2C). *Usp22* overexpression was confirmed in both epithelial cells and the surrounding stromal cells by immunoblots and by immunostaining of mammary gland sections (Figure 2D,E).

We next attempted to determine whether *Usp22* OE in the mammary gland epithelia is sufficient to induce the branching phenotype in vivo by introducing an *MMTV-Cre* transgene into mice bearing the *Usp22 LSL* allele (Appendix A).

Although we obtained heterozygous *LSL/+*; *MMTV-Cre/+* mice at the expected frequency, no homozygous *LSL/LSL*; *MMTV-Cre/+* mice were obtained (data not shown), suggesting that the *MMTV-Cre* transgene may be located on the same chromosome as the *ROSA26* locus. Since we observed increased branching in *OE/+* mice, we compared branching in mammary glands from 16-week old wild-type *MMTV-Cre/+* (without the *LSL* allele) and *LSL/+; MMTV-Cre/+* mice (Appendix A). We again observed an increase in the mean number of branching nodes in mice bearing the *Usp22* overexpressing allele (Appendix A). These results indicate that *Usp22* overexpression in mammary epithelia is sufficient to induce over-branching. However, our data do not rule out potential effects from the surrounding stroma or from hormonal stimulation that may arise upon ubiquitous *Usp22* overexpression.

### 2.3. Altered Gene Expression Profiles in Usp22 OE Primary Mammary Epithelial Cells

To determine whether the pronounced mammary gland phenotype we observed in the *Usp22*
*OE/OE* mice might reflect normal functions of endogenous *Usp22*, we analyzed *Usp22* expression in mammary epithelial cells (MECs) from wild type mice. We took advantage of publicly available single-cell RNA sequencing (scRNA-seq) data from MECs isolated at various stages of development, which previously identified mammary gland cellular subtypes [17]. In these studies, transcriptomes at single cell resolution were generated from Epcam+ MECs derived from nulliparous, mid-gestation, lactation, and post-involution adult mouse mammary tissues. These analyses identified 15 clusters representing different transcriptional cell states (Appendix A). We assessed the abundance of *Usp22* transcripts within each cluster (Appendix A) and found that *Usp22* is most ubiquitously expressed in luminal clusters, and this expression is most pronounced in hormone-sensing progenitors (clusters C1, C2), differentiated hormone-sensing cells (clusters C4, C5, C6), and alveolar cells (clusters C8, C10) that are only present in pregnant females, implicating *Usp22* in the regulation of mammary gland hormonal response (Appendix A).

We next isolated primary MECs and stromal fibroblasts from 3.5-month-old WT, *Usp22 OE/+*, or *Usp22 OE/OE* female mammary glands for transcriptomic analyses. Marker analyses for the mammary epithelial cells (CDK8 and Ecad) and the stromal fibroblasts (Fsp-1) confirmed the successful isolation of these cell populations (Appendix A).

We isolated RNA from MECs derived from seven pairs of littermate-matched *WT* and *Usp22 OE/OE* mice and performed RNA-seq to define differential expression gene signatures associated with *Usp22* overexpression. In total, we identified 361 upregulated genes and 277 downregulated genes upon the overexpression of *Usp22*. Significantly upregulated genes highlighted in the volcano plot (Figure 3A), including *Setd5, Pdk4, Bmpr1b, Esrrb*, and downregulated genes (e.g., *Kcne3*) were also verified by qRT-PCR (Figure 3B).

Gene ontology (GO) terms for physiological functions and diseases enriched for genes that were downregulated in *Usp22* OE MECs were related to inflammatory response, recruitment of white blood cells (myeloid cells and lymphocytes), endocrine gland tumor, and carcinogenesis at large (Figure 3C,G). In contrast, processes related to senescence and autophagy were consistently upregulated (Figure 3C,F). The GO terms for molecular and cellular functions affected by *Usp22* OE were determined using Ingenuity Pathway Analysis (IPA) software (Qiagen; www.ingenuity.com/products/ipa, accessed on 12 August 2020). Processes highlighted by terms enriched for upregulated genes upon USP22OE included the formation of focal adhesions and cellular protrusions, mitotic spindles, and the p53 pathway (Figure 3B,D). Several canonical pathways were identified as being induced, including integrin signaling, estrogen receptor signaling, ERK/MAPK signaling, as well as TGF-β signaling (Figure 3D,E). Interestingly, signaling pathways driven by IL-6/STAT3 and IL-2/STAT5 were both inhibited (Figure 3C,E).

We also performed an extensive gene set enrichment analysis (GSEA) that verified most of the GOs identified by IPA. Expression of individual genes in the top upregulated pathways identified by IPA and GSEA analysis was further examined by qRT-PCR using both primary cells (WT and *Usp22 OE* MECs) and immortalized normal NAMRU murine mammary gland (NMuMG) cells with and without *Usp22* overexpression (Appendix A). These data are again consistent with the upregulation of these pathways, although changes in the expression of some individual pathway component genes were minor.

### 2.4. Deregulation of Signaling Cascades in Primary Mammary Epithelial Cells Overexpressing Usp22

In our previous study, we identified multiple cytoplasmic signal transducers to be deregulated in mouse placentas upon deletion of *Usp22*, including components of multiple receptor tyrosine kinase pathways [15]. To determine whether changes in signaling effectors also occur when *Usp22* is overexpressed in the mammary gland, as indicated by our RNA-seq data, we performed immunoblots for AKT, ERK, GSK3β and their phosphorylated isoforms (p-AKT, p-ERK, p- GSK3β) in primary WT and *Usp22 OE* MECs (Figure 4A). Levels of p-AKT, p-ERK, and p-GSK3β were clearly elevated in the *Usp22 OE* cells, relative to the total protein levels. Levels of GRB2 were also increased, but E-CADHERIN levels were not (Figure 4A). These results indicate an upregulation of ERK/MAPK signaling components upon *Usp22* OE.

The over-branching phenotype may reflect increased proliferation or migration of *Usp22* OE cells. Interestingly, in induced endothelial cells derived from *Usp22* null embryonic stem cells, both decreased proliferative capacity and migration defects were observed [15]. To determine whether such cellular effects could be induced by *Usp22* OE in vitro, we used a retroviral *Usp22* OE vector (MSCV-Usp22-Puro), to infect an immortalized NMuMG cells (Figure 4B). *Usp22* OE showed a limited increase in the proliferation capacity of this mammary gland cell line (Figure 4C). Transwell migration assays also indicated that *Usp22* OE promotes the migration of these epithelial cells (Figure 4D,E). Increased phosphorylation of ERK in *Usp22* OE NMuMG cells was again observed, but phosphorylation of AKT was not increased, perhaps reflecting differences in intercellular communication in cells derived from mouse tissue compared to cultured human cell lines (Appendix A). Interestingly, the levels of H2B ubiquitination were unaffected by *Usp22* OE in NMuMG cells (Appendix A). We conclude that *Usp22*-dependent deregulation of signaling responses in mammary epithelial cells impact their proliferative and migratory behavior, consistent with the abnormal mammary gland morphogenesis we observed.

### 2.5. No Increase in Spontaneous Tumor Formation upon Ubiquitous Overexpression of Usp22 in Mice

To address whether overexpression of *Usp22* is sufficient to induce tumor formation in mice, we examined cohorts of WT, OE/+, and OE/OE mice for at least 2 years. No significant differences in survival of the cohorts were observed (Figure 5A, Appendix A). We selected three 2+-year-old mice per gender and genotype for full necropsy, including gross anatomical examinations and sectioning (H&E staining) of major organs. There was no increase observed in the incidence of tumors in the OE/+ or the OE/OE mice relative to the WT controls (Figure 5B and data not shown). Our data demonstrate that *Usp22* overexpression alone is not sufficient to promote tumorigenesis in mice.

## 3. Discussion

Our findings reveal that *Usp22* overexpression impacts branching morphogenesis of mammary glands in nulliparous female mice. Consistent with this phenotype, transcriptomic analyses indicate upregulation of key pathways involved in mammary gland branching in ductal epithelial cells derived from the *Usp22* overexpressing mice, including estrogen receptor, ERK/MAPK, and TGFβ signaling.

Branching morphogenesis is observed in tissues that need a rapid and dense expansion of the surface area for transport of materials between cells and tissues, such as the lungs, vascular networks, and mammary glands [18,19]. Although our study focuses on the prominent branching aberrations observed in the mammary gland, it is possible that *Usp22* overexpression affects the branching patterns of other organs, and we are currently investigating this possibility.

The branched structures of the mammary gland are regulated by multiple signaling cascades between the epithelia and the stroma, including FGFs, BMPs, WNTs, and EGFs [20,21,22,23,24,25,26]. The cooperation between growth factor cascades, integrin signaling pathways, and ovarian hormones is necessary to produce a fully functional mammary gland.

The mammary gland over-branching phenotype in the *Usp22* OE females again highlights connections between *Usp22* and signaling pathways [15]. Our morphological analyses indicate a specific impact of *Usp22* overexpression in bifurcation and arborization by secondary and side branching, but not a significant difference in the primary elongation of the mammary duct along the fat pad. The observed phenotype on nulliparous adult females (16 weeks of age) shows that *Usp22* overexpression creates a phenotype similar to the induced arborization of the mammary gland observed at the onset of pregnancy. This aberrant induction of arborization could be due to the aberrant activation of ER or PR signaling when *Usp22* is overexpressed. Further characterization will be required both at the cellular and the molecular level to understand the impact of *Usp22* overexpression on estrogen and progesterone responses. Interestingly, both *Usp22* and ATXN7L3, a *Usp22* interacting partner required for its enzymatic activity, are shown to be co-recruited with ER to different promoters and regulate the expression levels of downstream targets [27,28,29] in cancer cell lines.

Hyperactivation of AKT, ERK, and GSK3β has been associated with increased branching phenotypes in the post-pubescent mammary gland, as a result of deregulated signaling cascades like insulin like growth factor (IGF-1), estrogen receptor alpha and TGF β1 and EGF receptor, consistent with our results both in primary mammary epithelial cells and in mammary epithelial cell lines [30]. Proliferation is also required for ductal branching, and our data show upregulation of mitotic spindle regulators; however, we were not able to detect any increased proliferation of the mammary epithelial cells in situ, in contrast to previous reports of observed “hyperplasias” of the mammary gland associated with *Usp22* overexpression [13]. Our findings indicate that the increased branching morphogenesis is probably due to the deregulation of different signaling pathways (including TGFβ and RTKs) that are responsible for the regulation of the collective migration of mammary epithelial cells.

Similar to our loss of function analysis of *Usp22* in mouse embryogenesis, our genome-wide transcriptional profiling of the primary mammary epithelial cells and the observed deregulation of signaling molecules indicates that *Usp22* may regulate these pathways both transcriptionally and post-transcriptionally. *Usp22* has both histone and non-histone substrates, and it also functions in non-transcription-related processes [31,32], so the phenotypes we observe in the mammary gland may reflect a combination of gene expression changes and changes in the levels or activities of non-histone proteins that are regulated directly by *Usp22*-mediated deubiquitination. Interestingly, *Usp22* overexpression did not affect the levels of SAGA components such as GCN5 and ATX7L3, raising intriguing questions about its potential SAGA-independent functional activity. It is also possible that the overexpression of *Usp22* may sequester ATXN7L3 and ENY2 away from USP27x and USP51, thereby impacting the activities of these DUBs [33]. A systematic analysis of the protein levels of all the subunits of the DUB module in different human cancers is necessary in order to understand the molecular impact of *Usp22* overexpression in tumorigenesis.

Understanding the mechanisms that govern normal mammary gland development is crucial to the comprehension of breast cancer etiology. The role of *Usp22* in breast cancer has mostly been analyzed in cell lines, and a thorough analysis of *Usp22* in mouse models of breast cancer is currently lacking. Kim et al. (2017) identified c-MYC as a direct target of *Usp22* deubiquitinating activity and reasoned that *Usp22* overexpression stimulated breast cancer cell line growth and colony formation through the increased tumorigenic activity of c-MYC [34]. A second study identified *Usp22*-dependent deubiquitination and stabilization of ERa in MCF7 cells and their co-recruitment at promoters of genes regulated by ERa such as c-MYC and GREB1 [29]. The estrogen receptor pathway was found to be upregulated in our transcriptomic analysis; however, we were unable to observe any change in the levels of c-MYC protein in tissues or NMuMG cells overexpressing *Usp22*. While this manuscript was in preparation, a new study using breast cancer mouse models and human cell lines showed that *Usp22* is necessary for the oncogenic function of HER2 and identified the deregulation of unfolded protein response (UPR) as the potential underlying mechanism [35].

In our transcriptomic analyses, SETD5 was the most highly upregulated gene. Interestingly, SETD5 has been implicated in pancreatic cancer therapy resistance [36]. Although the role of SETD5 in breast cancer is largely unexplored, it is possible that its upregulation by the overexpression of *Usp22* may play a role in promoting aggressive cancer phenotypes.

*Usp22* overexpression was not sufficient to increase tumorigenesis in any mouse tissue. Our findings indicate that although *Usp22* may not function on its own as an oncogene, it may enhance signaling abnormalities associated with oncogenesis. Given the functional connections between the signaling pathways identified here and in previous studies by our lab and others, it will be of great interest to determine whether *Usp22* is required in tumors driven by those signaling cascades.

## 4. Methods

### 4.1. Generation of the Targeting Construct

The shuttle vector RfNLIII (a generous gift from Dr. Ming-Jer Tsai) was used to add a Flag-myc tag to the *N*-terminal of mouse *Usp22* cDNA. The base vector pCAGGS-LSL-luciferase (also a gift from Dr. Ming-Jer Tsai) was used as the parental vector for targeting construct [37]. To generate the pCAGGS-LSL-Usp22 targeting construct, mouse *Usp22* full cDNA was amplified by PCR and inserted into the SpeI site of the RfNLIII vector with correct orientation verified by enzyme digestion and DNA sequencing analysis. The backbone of this shuttle vector was removed through Sac I and Kpn I double enzyme digestion. As the remaining DNA fragment and the base vector shared DNA fragments, homologous recombination occurred through recombineering technology (https://frederick.cancer.gov/Science/BrbRepository/#/protocol, accessed on 12 January 2014) when they were transformed into SW102 cells by electroporation. Transformed hosts were cultured on LB plates at 30 °C for 48–60 h in the presence of 50 μg/mL ampicillin and 25 μg/mL kanamycin. The successfully and correctly recombined construct was transformed into 293-Flp cells for removal of the FRT-flanked kanamycin-resistant cassette to obtain the targeting construct. To confirm the correct targeting, the construct was sequenced twice through the cDNA and junctions of the components.

### 4.2. Gene Targeting in ES Cells and Experimental Mice

Animals were kept in regulated facilities, monitored daily, and all procedures that involved animal handling were performed in accordance with the approved Institutional Animal Care and Use Committee (IACUC) protocols at the University of Texas MD Anderson Cancer Center.

The targeting construct DNA described above (25 μg) was linearized by Pac I and electroporated into R1 embryonic stem (ES) cells conducted at Baylor College of Medicine. Correctly targeted ES clones were screened by puromycin for positive selection and diphtheria toxin A-mediated elimination of random integration for negative selection. Three targeted ES clones were identified by Southern blot as described in the Supplementary Material. In brief, genomic DNA of the ES clones was digested with EcoRV and subjected to Southern blotting. With a 5′ probe, the correct clones should yield an 11.5-kb hybridization band corresponding to the wild-type locus and a 4.5-kb band that represents the targeted ROSA26 locus. With a 3′ probe, the correct clones should yield an 11.5-kb hybridization band corresponding to the wild-type locus and a 9.5-kb band that represents the targeted ROSA26 locus. The production of chimeras from the ES cells was conducted at Baylor College of Medicine. Germline transmission of the *Usp22* allele was determined by Southern blotting and PCR assay. The allele-specific primers for PCR genotyping are 5-AAAGTCCCTATTGGCGTTACTA-3 (forward) and 5-AAAGTCGCTCTGAGTTGTTATC-3 (reverse) for the knock-in allele with a product size of 388 bp. PCR primers for the wild-type ROSA26 locus are 5-GGAGCGGGAGAAATGGATAT-3 (forward) and 5-AAAGTCGCTCTGAGTTGTTATC-3 (reverse) with a product size of 602 bp. The PCR program ran 35 cycles on 95 °C, 5 min, for denaturing; 55 °C, 40 s, for annealing; and 72 °C, 1 min, for extension. To obtain PCR genotyping products across homologous arms, a reverse primer 5-GGCTCCTCAGAGAGCCTCGG-3, together with two forward primers mentioned above, resulted in 1564 bp and 1342 bp products for wild-type and knock-in alleles, respectively.

To generate ubiquitous *Usp22* overexpression OE mice, we crossed Rosa26KI-Usp22-LSL mice with Zp3-cre strain (obtained from the Genetically Engineered Mouse Facility at MD Anderson Cancer Center) [16]. For the rescue experiment, we crossed the *Usp22* OE heterozygous and homozygous mice with heterozygous mice carrying the *Usp22* RRS377 allele described by Koutelou et al. [15]. Finally, we used the MMTV- cre line D mice (#003553, Jackson Laboratory, Bar Harbor, ME, USA) to generate mammary gland-specific *Usp22* overexpression mice.

### 4.3. Preparation of NMuMG Cells Stably Expressing Usp22 and Proliferation Assay

The NMuMG cells and the Phoenix ecotropic retrovirus packaging cell line (gift from Dr. Kai Ge) were cultured in DMEM medium containing 10% fetal bovine serum (FBS), penicillin, and streptomycin. *Usp22* cDNA was inserted into the pMSCV-puro vector (gift from Dr. Kai Ge). Control and *Usp22* expressing retroviruses were produced as described with modifications [38]. Briefly, Phoenix cells were seeded 24 h, then transfected with retrovirus vector by lipofectamine 2000 reagent (Invitrogen, Waltham, MA, USA). Twelve hours after transfection, the cells were changed to fresh medium. Virus-containing supernatant was collected at 24 h, filtered through a 0.45-µm membrane, and supplemented with 8 µg/mL polybrene (Sigma, St. Louis, MO, USA). NMuMG cells were incubated with 1:1 media-diluted retrovirus for 36 h, then 1:3 split or 1:4 before selection with 2 µg/mL puromycin (Sigma) for 5 days. For the proliferation assay, control and *Usp22*-expressing NMuMG cells were cultured in DMEM medium containing 1% FBS overnight, after which 2000 cells were seeded in complete medium in 24-well plates. Cells were trypsinized and cell numbers were counted at 4, 6, 8, 10, and 12 days using a trypan blue exclusion assay.

### 4.4. Isolation and Confirmation of Mouse Primary Mammary Gland Epithelial and Fibroblast Cells

Left and right number 2, 3, and 4 mammary glands from three-and-a-half-month-old *Usp22*-overexpressing and wildtype mice were dissected with lymph nodes removed from number 4 glands. Organoids and fibroblasts were isolated using the method described [39] with modification. Briefly, the glands were minced into a paste and incubated in DME/F-12 (HyClone, Logan, UT, USA) with 2 mg/mL collagenase A (Roche, Basel, Switzerland) and 100 units/mL hyaluronidase (Sigma-Aldrich, H3506) for 1 h at 37 °C with 180 rpm rotation along with manual shaking every 15 min. The cells were washed with DME/F-12 and centrifuged twice at 450 g for 10 min to remove the fatty layer. The cells were incubated in DME/F-12 with 2 units/mL DNase I (Sigma-Aldrich, D2463) at room temperature for 3 min and centrifuged at 450 g for 10 min. Then, the pellets were resuspended in red blood cell lysis buffer (Stemcell Technology, Vancouver, BC, Canada) on ice for 2 min and washed with DME/F-12 at 450 g for 10 min. Differential centrifugation (pulse centrifugation to 450 g for 10 s) was used to separate stroma from organoids. The supernatant from two spins containing the fibroblasts, macrophage, and endothelial cells was pelleted and used as stroma fraction. To obtain fibroblasts, the pellets were resuspended in DME/F-12 containing 10% FBS on a 10 cm tissue culture plate. After plating the resuspended cells for 45 min, the plates were washed 3 times with PBS and attached cells were collected as fibroblasts. The pellets from 2 round pulse centrifugations were organoids, mainly consisting of mammary gland epithelial cells. The organoids were digested with 0.05% Trypsin/EDTA (HyClone) with 2 unit/mL of DNase I for 3 min and an equal volume of DME/F-12 containing 10% FBS was added, and the cells were pipetted 20 times and filtered through 40 μm strainer. After 3 min at 600 g centrifugation, single epithelial cells were obtained for experimental use. For validation of the purity of the isolation, freshly isolated epithelial cells were plated in mammary epithelium basal medium (MEBM, from Lonza) for 48–72 h and fibroblasts were plated in DME/F-12 containing 10% FBS before collection for Western blot, qRT-PCR or immunofluorescence staining in Chamber slides using E-cadherin (antibody from Invitrogen) or Cytokeratin 8 (antibody from *Developmental Studies Hybridoma Bank* (DSHB)) as an epithelial marker and FSP-1 (antibody from Proteintech) as a fibroblast marker.

### 4.5. Western Blotting

Cells or tissue samples were lysed in RIPA buffer with protease inhibitor cocktail (Sigma) and phosphatase inhibitors PhosSTOP (Roche). The extracts were dissolved in Laemmli buffer and boiled at 95 °C for 5 min. The lysates (20–30 μg of total protein) were separated with SDS-PAGE gels, transferred to PVDF membrane (BIO-RAD) then blotted with primary and secondary antibodies. The primary antibodies used were anti-Usp22 antibody (Abcam or homemade as described previously) [15], Flag-tag antibody (Sigma or Cell Signaling Technology), Myc-tag antibody (Santa Cruz), Gapdh antibody (Millipore, Burlington, MA, USA), FSP-1 (Proteintech, Rosemont, IL, USA), H2Bub (Millipore) and cytokeratin 8 (Developmental Studies Hybridoma Bank). Total Akt antibody and phosphor-Akt(S473), total p44/42 (ERK1/2), phosphor-p44/42, GCN5, phosphor-GSK3β, GSK3β, GRB2, E-cadherin and H2B antibodies were from Cell Signaling Technology. ATXN7L3 was a gift from Dr. Tora [40].

### 4.6. Immunoprecipitation

For FLAG IPs, the procedure was described previously [33]. Anti−FLAG beads (M2, Sigma) were incubated with tissue or cell lysates for 4 h at 4 °C on a rocking platform, harvested by centrifugation, washed once in IP buffer containing 100 mM NaCl, two times in IP buffer containing 400 mM NaCl, once in IP buffer containing 200 mM NaCl, and once in TBS. The beads were then incubated with 100 μg/mL (*v*/*w*) 3xFLAG (Sigma) peptide in TBS for 20 min at 4 °C on a rocking platform to elute the precipitated complexes. The clear eluates were mixed with SDS sample buffer and resolved by SDS-PAGE.

### 4.7. RNA Isolation and qRT-PCR

Mouse tissues or mammary gland epithelial cells were isolated as described above. Total RNA from cell pellets was extracted using RNeasy Plus Mini Kit (Qiagen, Hilden, Germany). The extracted RNA samples were treated with RNase-free DNase set (Qiagen). qRT-PCRs were measured using the SYBR Green method using the 7500 Fast Real-Time PCR System (Applied Biosystems, Waltham, MA, USA). The sequences for the gene-specific qRT-PCR primers used are listed below. The relative expression of RNAs was calculated using the comparative Ct method with *Gapdh* as an internal control. Data are presented as normalized individual data points.

### 4.8. Mammary Gland Whole-Mount Staining

Number 4 mammary glands were dissected, spread out on slides, fixed in 10% formalin in PBS overnight, and kept in 70% ethanol for 24 h. The glands were soaked in 35% and 17.5% ethanol for 1 h each. The glands were rinsed with PBS and stained in carmine alum (contains carmine and aluminum potassium sulfate in water) overnight. Next, the glands were dehydrated in a series of ethanol solutions for 1 h each. The glands were cleared in xylenes overnight and mounted with a coverslip using permount. A dissecting microscope was used to image the carmine-stained glands. Branching was quantified by counting the total number of branch nodes found in about ¼ of the branching area from the lymph node to the ending point and normalized as numbers of nodes per 6 × 10^6^ μm^2^ areas.

### 4.9. Mammary Gland Immunofluorescence Staining

For immunofluorescence staining, number 4 mammary glands were fixed in 10% formalin for 24 h, moved to 70% ethanol solution for 24 h, and embedded in paraffin. Then, 5 μm sections were cut and dehydrated in xylene and an ethanol gradient. After performing antigen retrieval in 10 mM citrate sodium containing 0.05% tween-20, the sections were washed with PBS and incubated with primary antibody overnight at 4 °C in antibody solution containing 10% donkey serum and 0.05% tween-20. Slides were washed three times in PBS containing 0.05% Triton X-100 and incubated with secondary antibody in antibody solution for 1 h at room temperature in the dark. Slides were washed in PBS, incubated with 4′, 6-diamidino-2-phenylindole (DAPI) for 5 min at room temperature, washed in PBS, mounted with coverslips using ProLong^®^ Gold antifade reagent (Life Technologies, Carlsbad, CA, USA). Slides were imaged on a Leica DMI 6000B fluorescence microscope. The primary antibodies used are Rat anti-Cytokeratin 8 (DSHB) and Rabbit anti-Flag-tag (Sigma). The secondary antibodies were conjugated with either Alexa fluor 488 or Alexa fluor 568 (Molecular probe).

### 4.10. Whole Transcriptome Sequencing (RNA-seq)

For the experiment, total RNA from mammary gland epithelial cells of seven pairs of littermate-matched 3.5-month wild-type and Rosa26KI-Usp22 OE/OE mice was extracted using the RNeasy Plus Mini Kit. The purified RNA was treated with RNase-free DNase set (Qiagen) and used for library preparation. Briefly, 1 µg of total RNA was used to prepare RNA-seq libraries with the Illumina TruSeq mRNA Kit according to the manufacturer’s protocol. The libraries were then sequenced using a 2 × 75 base paired-end protocol on the Illumina HiSeq 3000; 24–52 million pairs of reads were generated per sample. Each pair of reads represents a cDNA fragment from the library. The reads were mapped to the mouse genome (mm 10) using TopHat (version 2.0.10) [41]. By reads, the overall mapping rate is 93–97%; 89–95% of fragments have both ends mapped to the mouse genome. The number of fragments in each known gene from GENCODE Release M21 [42] was enumerated using htseq-count from the HTSeq package (version 0.6.0) [43]. Only genes with at least 10 fragments in at least 7 samples were retained for differential expression analysis. The differentially expressed genes (DEGs) were identified by paired t-test using normalized read counts in genes estimated by DESeq (version 1.18.0) [44]. Although the purity of MECs from all 7 pairs of mice was similar, as determined by qRT-PCR for epithelial and stromal markers, samples originating from different mice showed a high degree of variability, making it difficult to identify DEGs, and thus relaxed criteria (*p*-value ≤ 0.05 and gene length > 200 bp) were used to identify genes as being differentially expressed. The IPA analysis was performed through the use of IPA [45] QIAGEN Inc. (https://www.qiagenbioinformatics.com/products/ingenuity-pathway-analysis, (accessed on 12 August 2020). The DEGs were provided to IPA with log2 ratios to show the expression change directions. IPA “Core Analysis” was run against its “Ingenuity Knowledge Bases (Genes Only)” to look for enriched pathways and functions. All the other parameters were set as default. The gene set enrichment analysis was performed by GSEA software [46]. The genes were pre-ranked by a signal-to-noise score (defined as the mean of log2ratio values from all 7 pairs of samples divided by the standard deviation of log2ratio values) and all the other parameters were set as default. Volcano plots comparing gene expression fold change between *Usp22* overexpressing cells and wild-type cells were generated using VolcaNoseR [47]. *Usp22* expression during mammary development was plotted using the web app at https://marionilab.cruk.cam.ac.uk/mammaryGland/ (accessed on 30 June 2020), which allows interactive exploration of scRNA data published in Bach et al. [17].

### 4.11. Transwell Migration Assays

NMuMG control and *Usp22* expressing cells were seeded and replaced with DMEM medium containing 1% FBS for 24 h. The cells were washed with PBS, trypsinized for 10 min, washed, and resuspended in DMEM medium containing 1% BSA. A total of 30,000 cells per well were plated onto 8 μm-pore transwell filters (BD Transduction Laboratories) into 24-well plates with 800 μL of DMEM media containing 10% FBS added to each well below the filter. The cells were allowed to migrate through the filter for 24 h, at which time the medium was removed and the upper surface of the filter was scraped twice with a cotton swab to remove any cells that did not migrate through the filter. The filters were fixed in 100% methanol for 5 min and stained with 0.5% crystal violet for 2 h. The filters were washed with dH_2_O and air-dried at room temperature. The migrated cells on the transwells were observed and images were taken using a ZEISS Stemi 2000-C dissecting microscope. Then, 250 µL of 10% acetic acid per sample was added into the under well and shaken for 15 min to dissolve the crystal violet. A total of 150 µLof solution per sample was used with a regular 96 well plate to measure the absorbance value (λ = 595 nm) using an Omega plate reader.

## 5. Conclusions

Our results further demonstrate that *Usp22* is important for proper signaling through multiple pathways that impact both normal development and oncogenesis. They also demonstrate that *Usp22* OE alone is not oncogenic, although it may facilitate tumor formation induced by oncoproteins that drive aberrant signaling.

## Figures and Tables

**Figure 1 cancers-13-04276-f001:**
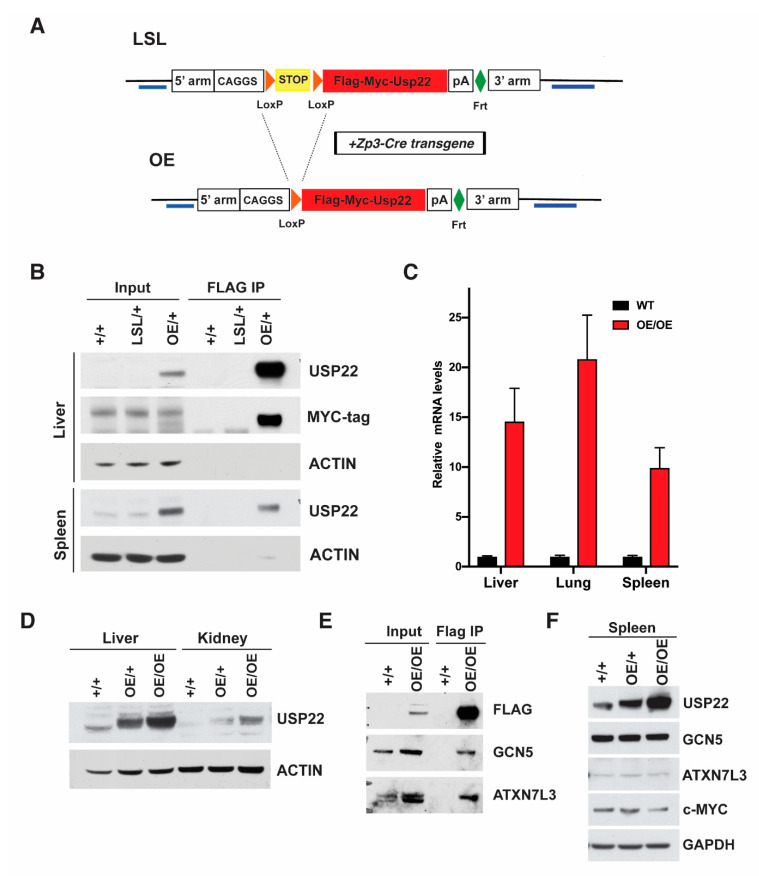
Generation of a mouse model that conditionally overexpresses *Usp22* transgene. (**A**) Female Rosa26KI-Usp22-LSL mice were crossed with male Zp3-cre mice, following the mating scheme described in Lewandoski et al., 1997, in order to acquire progeny with ubiquitous expression of *Usp22* transgene under the Zp3 promoter. Blue bars show the locations of 5′ and 3′ probes used for the Southern blot analysis. (**B**) Immunoprecipitation with anti-FLAG antibody using proteins from whole cell extractions of liver or spleen followed by immunoblotting with anti-MYC tag and anti-Usp22 antibodies to confirm the expression of the tagged *Usp22* protein in *OE/+* mice, but not in LSL/+ mice. (**C**) qRT-PCR analysis of total RNA isolated from the liver, lung, and spleen of wild type or *Usp22 OE/OE* adult female mice to validate the overexpression of *Usp22*. (**D**) Representative immunoblots showing the increased expression of *Usp22* in the liver and kidney of OE/OE mice relative to their wild type and heterozygous littermates. (**E**) Immunoprecipitation with anti-FLAG antibody using whole cell extracts from liver followed by immunoblotting with anti-GCN5 and anti-ATX7L3 antibodies to confirm the association of FLAG-Usp22 with both the DUB and HAT modules of SAGA in OE/OE mice. (**F**) Representative immunoblots using whole cell extracts from the spleen of wild type, *Usp22* OE/+, or *Usp22 OE/OE* adult mice showed that the levels of endogenous GCN5, ATXN7L3, and c-MYC proteins were not affected by *Usp22* overexpression.

**Figure 2 cancers-13-04276-f002:**
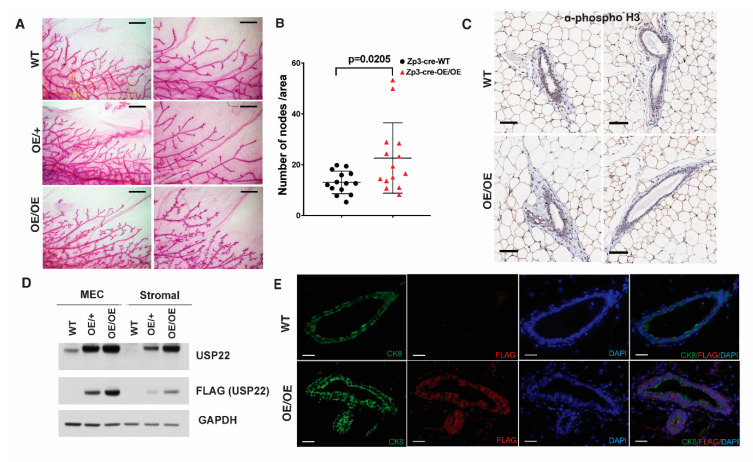
*Usp22* ubiquitous overexpression induces mammary gland over branching in post-pubertal nulliparous mice. (**A**) Mammary glands from nulliparous females at 16 weeks were stained with carmine alum. Representative images of mammary gland trees from wild type, heterozygous OE/+, and homozygous OE/OE female mice are shown. Scale bar—2 mm. (**B**) Quantification of mammary gland branching by counting the total number of branch nodes found in ¼ of the branching area from the lymph node to the ending point in each genotype (n = 14) and normalized as numbers of nodes per 6 × 10^6^ μm^2^ areas. *p* value was calculated with unpaired t test. (**C**) Anti-phospho histone H3 immunostaining of mammary gland paraffin sections from 16-week-old wild type and OE/OE nulliparous females show no overt increase in cell proliferation. Scale bar—150 µm. (**D**) Representative immunoblots using whole cell extracts from primary MECs and stromal fibroblasts from 3.5-month-old wild type, *Usp22 OE/+* or *Usp22 OE/OE* female mammary glands to validate the overexpression of *Usp22*. (**E**) Immunostaining of mammary gland paraffin sections with anti-FLAG and anti-CK8 antibodies confirm the expression of *Usp22* in both epithelial and stromal cells. Scale bar—50 µm.

**Figure 3 cancers-13-04276-f003:**
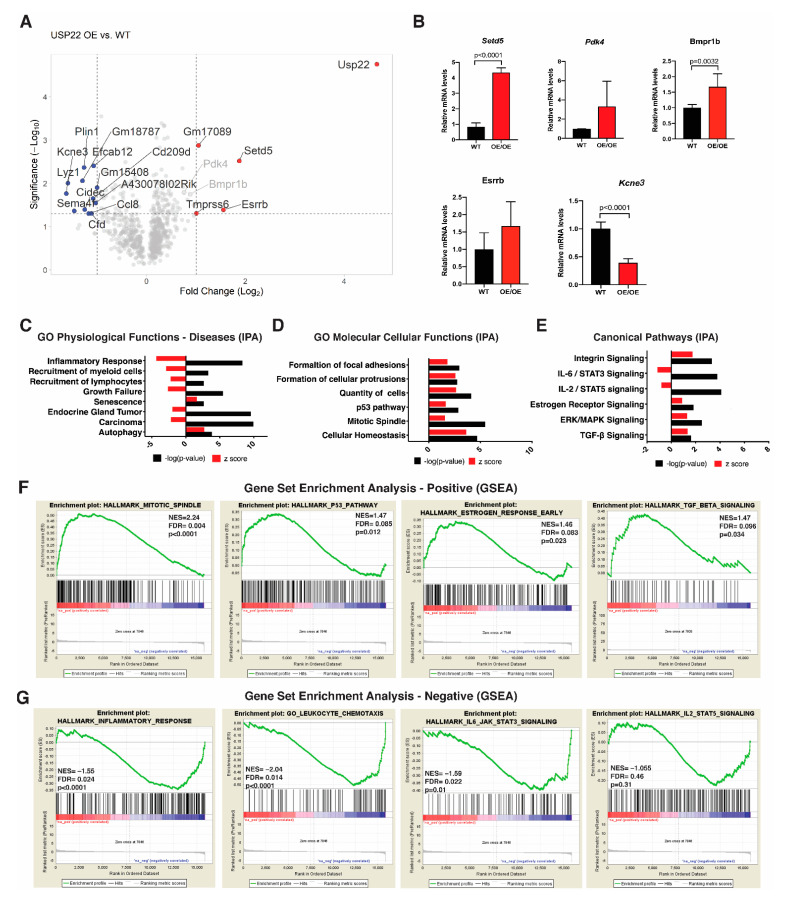
Altered gene expression profiles in *Usp22* overexpression primary mammary epithelial cells identify deregulated signaling cascades. (**A**) Volcano plot representation of differential expression analysis of genes upon *Usp22* overexpression. Differential expression of 638 genes reached statistical significance, with 361 up-regulated and 277 down-regulated genes, with the fold-change threshold set at |1| and *p*-value threshold set at *p* < 0.05. Red dots represent the top upregulated genes including *Usp22*, Setd5, and Esrrb, and blue dots represent the top downregulated genes including Kcne3, Sema6f, and Ccl8. (**B**) qRT-PCR analysis of total RNA from primary MECs isolated from 3.5-month-old wild type or *Usp22 OE/OE* female mammary glands to validate the top up-regulated and down-regulated genes identified. Shown is the relative expression of the genes normalized to Gapdh using three replicates per genotype. *p* values were calculated with unpaired t test and error bars depict SD of the three replicates. (**C**) GO analysis of significantly affected physiological functions and diseases predicted by IPA. (**D**) GO analysis of significantly affected molecular and cellular functions predicted by IPA. (**E**) Upstream regulators analysis by IPA predicted the IL6/STAT3 and IL2/STA5 pathway to be the most significantly inhibited, whereas integrin, estrogen receptor, ERK/MAPK, and TGF-β signaling pathways were predicted to be activated in the MECs overexpressing *Usp22*. (**F**,**G**) Representative plots of gene set enrichment GSEA analysis gene signatures that are positively (**F**) or negatively (**G**) affected by *Usp22* overexpression. FDR, false discovery rate; NES, normalized enrichment score.

**Figure 4 cancers-13-04276-f004:**
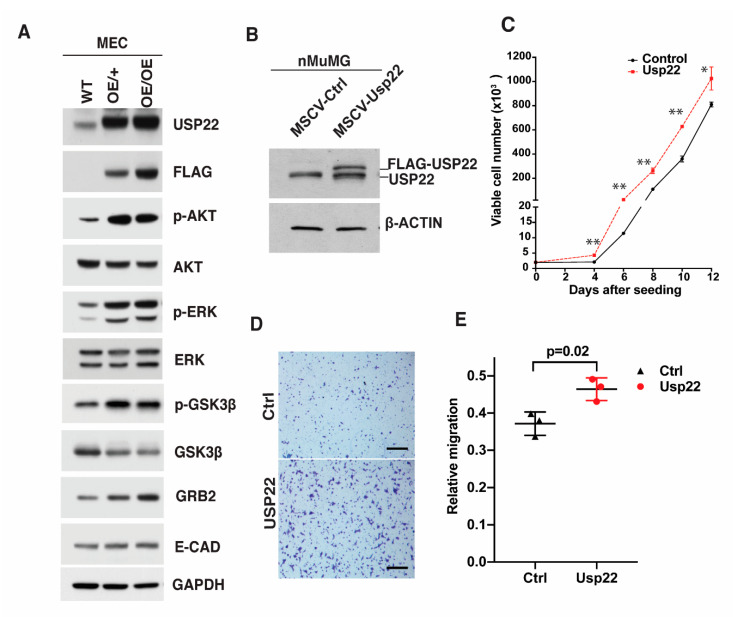
Deregulation of signaling cascades in primary epithelial cells and mammary cell lines upon *Usp22* overexpression. (**A**) Representative immunoblots showing increased levels of phosphorylated forms of ERK1/2, AKT, and GSK3β, as well as increased protein levels of GRB2, but not E-cadherin, in MECs overexpressing *Usp22*. (**B**) Immunoblot showing the stable overexpression of FLAG-tagged *Usp22* in immortalized NMuMG cells. (**C**) Growth assays indicate that *Usp22*-overexpressing NMuMG mouse epithelial cells show a limited increase in their proliferative capacity compared to the control cell line. ** *p* < 0.01, * *p* < 0.05. (**D**) Representative bright-field images of transwell migration assays using control or *Usp22* overexpressing NMuMG mouse epithelial cells, stained with crystal violet. Scale bar—400 µm. (**E**) Quantification of migrating *Usp22* overexpressing and control NMuMG mouse epithelial cells, measurements performed with crystal violet absorbance value of migrated cells from three replicates per group. Error bars represent SD of three replicates per group. *p* values were calculated using unpaired *t* tests.

**Figure 5 cancers-13-04276-f005:**
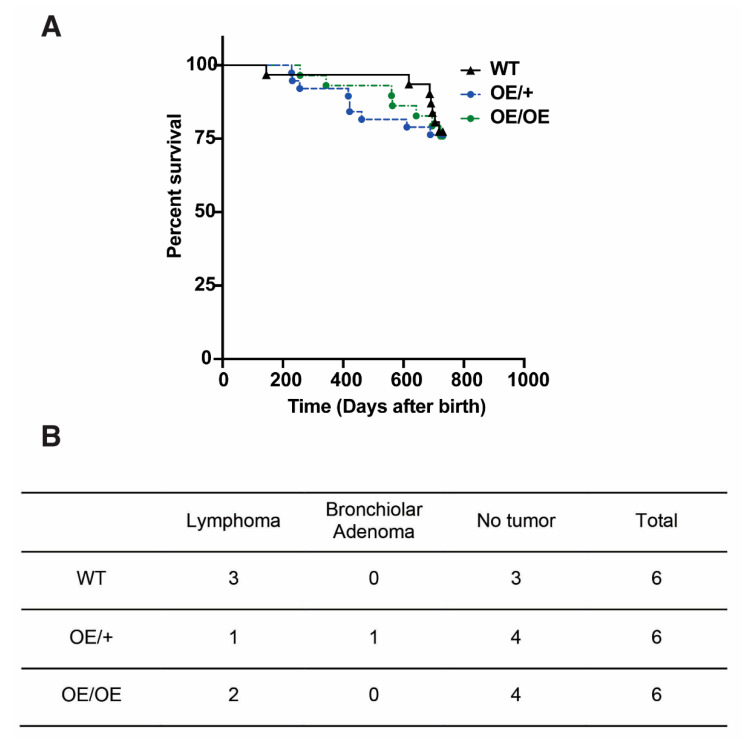
(**A**) Free survival of adult mice of wild type, *Usp22* OE/+ or *Usp22* OE/OE shows no significant differences between the different cohorts observed. (**B**) Necropsy results of 6 total 2+ year-old mice per genotype show no increased incidence of tumors in the OE/OE homozygotes or OE/+ heterozygotes compared to their wild type littermates.

**Table 1 cancers-13-04276-t001:** Mendelian ratios of the respective number of adult mice collected from the intercrossing of heterozygotes *Usp22* OE/+ are as expected.

	+/+	OE/+	OE/OE	Total
Observed number	45 (25%)	81 (46%)	52 (29%)	178 (100%)
Expected number	44 (25%)	89 (50%)	45 (25%)	178 (100%)
Chi square	0.23	0.72	1.09	
*p* value	>0.75	>0.25	>0.25	

**Table 2 cancers-13-04276-t002:** Mendelian ratios of the respective number of adult mice collected from OE/+ or OE/OE *Usp22* line crossed with *Usp22*^+/−^ heterozygotes for the *Usp22* null allele described in Koutelou et al., 2019.

	Total Number: 34			R26OE-Usp22: OE/+ Total Number: 108			R26OE-Usp22: OE/OE Total Number: 56		
	*Usp22* *+/+*	*Usp22* *+/-*	*Usp22* *-/-*	*Usp22* *+/+*	*Usp22* *+/-*	*Usp22* *-/-*	*Usp22* *+/+*	*Usp22* *+/-*	*Usp22* *-/-*
Observed number	12	22	0	32	56	20	18	22	16
Expected number	11	22	11	27	56	27	14	28	14
Chi square	0.09	0	11	0.93	0	1.81	1.14	1.29	0.29
*p* value	>0.75	0.99	<0.001	>0.25	0.99	>0.10	>0.25	>0.25	>0.50

## Data Availability

RNA-seq data have been deposited in GEO and the accession number is GSE180005.

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
