# Peer review of "Usp22 Overexpression Leads to Aberrant Signal Transduction of Cancer-Related Pathways but Is Not Sufficient to Drive Tumor Formation in Mice"

_cancers, 2021, doi:10.3390/cancers13174276_

Round 1
Reviewer 1 Report
USP22 expression is related to increased tumor growth, metastatic risk, and survival in prostatic cancers. Its overexpression is also observed in breast and pancreatic cancers.
Conditional overexpression of USP22 in mouse prostate increased proliferation of epithelial cells, but no neoplastic transformations were observed. In another study, USP22 showed tumor suppressor function in colorectal cancer, suggesting its role in cancer has not been established.
The authors generated unbiased mouse model with global overexpression of USP22, and found aberrant transduction of cancer-related pathways in over branching mammary glands. However, the changes are not sufficient to drive tumor-formation in mice.
McCann et al (Cancer Research 2020) argued that USP22 is an oncogenic driver in their previously published paper, although they did not observe any tumor formation in their conditional USP22-overexpressing transgenic mice, which may mislead the role of USP22 in cancer.
The present authors’ conclusion that USP22 is not sufficient to drive tumor formation from their study with USP22 unbiased over-expressing transgenic mouse model, is quite opposite to that by McCann et al. Therefore, this manuscript should be published ASAP.
The reference format may be corrected before publication.
Author Response
We would like to thank the reviewer for highlighting the importance of our study on USP22 functions in cancer. We do believe our findings that USP22 on its own does not have oncogenic function is critical for the field and needs increased attention.
We have corrected the reference format throughout the manuscript to match the current journal and we apologize for the oversight.

Reviewer 2 Report
This manuscript by Kuang et al. describes their genetic analyses of USP22 over expression and induction of pro-tumorigenic transcriptional and signaling using a novel inducible mouse model of Usp22 induction. USP22 is known to be highly expressed across different tumor types but its potential role in tumorigenesis is not well understood. This work builds on the former work of the research group, which demonstrated that Usp22 depletion perturbs TGF-beta and RTK signaling in development. In this manuscript, the authors investigate the potential pathways controlled by USP22 upregulation in the absence of other genetic lesions, focusing on the mammary gland. Interestingly, the authors report an increased mammary duct branching phenotype in nulliparous females, associated with increased proliferation , transcriptional regulation of ER and TGFB pathways and enhanced signaling through RAS and AKT in the mammary epithelium of their mouse model. The authors also link upregulated USP22 with distinct cellular characteristics defined by publicly available single-cell RNASeq analysis of mammary epithelial cells. Usp22 upregulation seems to be relevant in luminal clusters and in cells with hormone-sensing expression signatures. Perhaps not unexpectedly, up regulation of Usp22 alone is insufficient to induce tumorigenesis in the mammary gland of this novel mouse model.
Overall, this study is well presented and the data support the premise of the study, with the conclusions supported by the data.
One obvious question is whether USP22 upregulation is directly responsible for the transcriptional and signaling changes observed. USP22 is part of the SAGA complex, which has been shown to have broad roles in transcription and post-translational regulation. The function of the SAGA complex in this study is only briefly addressed to show that the expression of two complex members is unchanged in response too USP22. It is not addressed whether the depletion of either GCN5 or ATXN7L3 can rescue the branching phenotype observed in the overexpression model. This is an interesting question that could address whether USP22 up regulation is directly linked to SAGA function.
There are a number of follow-up areas evident from these studies, including whether USP22 upregulation contributes to tumor initiation or progression in the breast or in other cell types, such as lung; or whether high USP22 occurs in response to therapy. Further, what are the cooperating loci with USP22 to drive tumorigenesis in this model?
While these are provocative questions, they are beyond the scope of this manuscript. Importantly, this inducible mouse model of Usp22 is a novel and valuable resource to further develop new cancer models to understand the role of Usp22 upregulation in breast and other cancer types, in combination with other cancer initiating alleles.
I support the acceptance of this manuscript.
Author Response
We greatly thank the reviewer for the careful and thorough evaluation of our manuscript. All the questions that are brought up are critical and we are interested in understanding both the role of Usp22 as an integral component of SAGA in branching morphogenesis, as well as the role of USP22 upregulation in different types of cancer. Our current efforts are focused on the role of USP22 in breast cancer and we are working on reporting our findings soon.
We have performed detailed and thorough editing of our manuscript and we have corrected all the typos and language mistakes we identified.
